# Quality versus emergency: How good were ventilation fittings produced by additive manufacturing to address shortages during the COVID19 pandemic?

Roman Hossein Khonsari[1,2,3]*, Mathilde Oranger[4,5], Pierre-Marc François[6], Alexis Mendoza-Ruiz[7], Karl Leroux[7], Ghilas Boussaid[4], Delphine Prieur[3], Jean-Pierre Hodge[7], Antoine Belle[8], Vincent Midler[9], Capucine Morelot-Panzini[5,10], Maxime Patout[5,10,11], Jésus Gonzalez-Bermejo[4,5,10]

1 Service de Chirurgie Maxillo-Faciale et Chirurgie Plastique, Hôpital Necker - Enfants Malades, Assistance Publique – Hôpitaux de Paris, Paris, France, 2 Faculté de Médecine, Université Paris Cité, Paris, France, 3 Délégation Inter-Départementale pour le Développement de la Fabrication Additive (DIDDFA), Direction générale, Assistance Publique – Hôpitaux de Paris, Paris, France, 4 Service de Réhabilitation Respiratoire (Département R3S), Hôpital Pitié-Salpêtrière, Assistance Publique – Hôpitaux de Paris, Paris, France, 5 Faculté de Médecine, Sorbonne Université, Paris, France, 6 BONE 3D, Paris, France, 7 ASV Santé, Gennevilliers, France, 8 Service de Pneumologie, Centre Hospitalier Intercommunal de Compiègne-Noyon, Compiègne, France, 9 Département de la Maîtrise d'Ouvrage et de la Politique Technique – DEFIP, Assistance Publique - Hôpitaux de Paris, Paris, France, 10 Neurophysiologie Respiratoire Expérimentale et Clinique, INSERM UMRS1158, Paris, France, 11 Service des Pathologies du Sommeil (Département R3S), Hôpital Pitié-Salpêtrière, Assistance Publique - Hôpitaux de Paris, Paris, France

* roman.khonsari@aphp.fr

**Data Availability Statement:** All relevant data are within the manuscript and its Supporting information files.

## Abstract

### Objective

The coronavirus disease pandemic (COVID-19) increased the risk of shortage in intensive care devices, including fittings with intentional leaks. 3D-printing has been used worldwide to produce missing devices. Here we provide key elements towards better quality control of 3D-printed ventilation fittings in a context of sanitary crisis.

### Material and methods

Five 3D-printed designs were assessed for non-intentional (junctional and parietal) and intentional leaks: 4 fittings 3D-printed in-house using FDeposition Modelling (FDM), 1 FDM 3D-printed fitting provided by an independent maker, and 2 fittings 3D-printed in-house using Polyjet technology. Five industrial models were included as controls. Two values of wall thickness and the use of coating were tested for in-house FDM-printed devices.

### Results

Industrial and Polyjet-printed fittings had no parietal and junctional leaks, and satisfactory intentional leaks. In-house FDM-printed fittings had constant parietal leaks without coating, but this post-treatment method was efficient in controlling parietal sealing, even in devices with thinner walls (0.7 mm vs 2.3 mm). Nevertheless, the use of coating systematically

**Funding:** The authors received no specific funding for this work.

**Competing interests:** PMF is the CTO of BONE 3D (Paris). AMR, KL, and JPH are employees of ASV Santé (Genevilliers, France). This does not alter our adherence to PLOS ONE policies on sharing data and materials.

induced absent or insufficient intentional leaks. Junctional leaks were constant with FDM-printed fittings but could be controlled using rubber junctions rather than usual rigid junctions. The properties of Polyjet-printed and FDM-printed fittings were stable over a period of 18 months.

## Conclusions

3D-printing is a valid technology to produce ventilation devices but requires care in the choice of printing methods, raw materials, and post-treatment procedures. Even in a context of sanitary crisis, devices produced outside hospitals should be used only after professional quality control, with precise data available on printing protocols. The mechanical properties of ventilation devices are crucial for efficient ventilation, avoiding rebreathing of $CO_2$, and preventing the dispersion of viral particles that can contaminate health professionals. Specific norms are still required to formalise quality control procedures for ventilation fittings, with the rise of 3D-printing initiatives and the perspective of new pandemics.

## Introduction

Non-Invasive Ventilation (NIV) and Continuous Positive Airway Pressure (CPAP) are standards of care for chronic hypercapnia and respiratory failure [1], sleep apnoea, and acute hypoxemia, such as in severe SARS-CoV-2 infections [2]. Several NIV and CPAP devices include fittings with intentional leaks, either on the ventilation mask itself, or between the mask and circuit, to prevent exhaled air to be rebreathed. Since the beginning of the COVID19 pandemic, exhaled particle dissemination has been documented [3] and emerged as an unexpected new problem with contamination risks for the immediate environment of ventilated patients.

To tackle this issue, non-ventilated masks have been proposed as a first-line option with the addition of an expelled expiration port (whisper swivel or similar) [3]. Antibacterial and viral filters interposed between the mask and the exhalation ports have also been proposed [3–6]. Thus, particle dispersion during ventilation could be controlled with sealed mask-interface connections [3].

During the pandemic, shortages in ventilation devices and specifically in fittings with intentional leaks occurred worldwide [7–12]. 3D-printing was extensively used to overcome shortages due to extraordinary needs and interruptions in supply chains, with little focus on quality control and risk management of medical devices requiring compliance to strict ISO standards and CE marking [13,14]. In fact, while 3D printing was successfully used as a versatile emergency solution in many centres in Europe and worldwide, showing its ability to act as a support solution during sanitary crises, the main issue recurrent issue in most reports was the lack of formal certification when medical devices were produced [14].

Within the trust of Greater Paris academic hospitals (Assistance Publique–Hôpitaux de Paris, AP-HP), the largest hospital trust in Europe grouping 39 hospitals, covering an area of over 10 million inhabitants, and employing nearly 100,000 people, a centralized 3D-printing initiative was launched in April 2020 with 60 professional Fused Deposition Modelling (FDM) 3D-printers (F120, F170 and F370, Stratasys, Eden Prairie, USA) and a team of 5 full-time engineers working 24/7 (BONE 3D, Paris). The aim of the project was to provide accelerated design and production services to all AP-HP employees facing various shortages due to the pandemic [15–17]. Over two hundred designs were produced from March to November 2020, with approximately 40,000 pieces printed and distributed within the AP-HP network. Among

these designs, the central platform produced various models of ventilation devices, using black Acrylonitrile Butadiene Styrene (ABS) M-30 (355–02112, Stratasys, Eden Prairie, USA) as a raw material. ABS is a widely used engineering thermoplastic with high durability, and printed ABS has up to 80% of the strength of injection-molded ABS, making it suitable for functional applications. ABS M-30 is characterized by its strength and toughness, while being lightweight and resilient: ultimate tensile strength 32 MPa, Izod impact strength (unnotched) 7%, and elongation at break 300 ohms (data provided by the manufacturer). In parallel, several hospitals within AP-HP had pre-pandemic local 3D-printing platforms and produced significant amounts of ventilation devices during the crisis–notably, Necker–Enfants Malades Hospital contributed to the COVID19 effort by printing ventilation fittings using Polyjet technology (J735, Stratasys, Eden Prairie, USA) with biocompatible transparent MED610 (Stratasys, Eden Prairie, USA) resin as a raw material. Finally, during the first wave of the pandemic, all AP-HP hospitals received generous daily deliveries of ventilation devices printed externally by independent makers owning private 3D-printers, or by various independent manufacturers, with little information on designs and production protocols, and thus insufficient quality control and difficult match with the real needs of clinical departments treating COVID19 patients [14].

The main objective of this study was to assess non-intentional and intentional leaks in a series of ventilation devices produced during the first wave of the pandemic and delivered to AP-HP clinical departments. We considered (1) in-house devices produced by the central AP-HP 3D-printing platform using FDM printing technique for which ABS was a raw material, (2) in-house devices produced by one academic AP-HP hospital using Polyjet printers and MED610 as a raw material, and (3) externally produced devices delivered to AP-HP by independent manufacturers. Our results suggest that in-house and external devices should be used only after professional quality control and that unsupervised 3D-printing of devices with intentional leaks can lead to harmful situations for patients and healthcare professionals. Rigorous approaches to quality control are furthermore mandatory steps to obtain certification for the 3D-printing of medical devices [18–20]. Finally, the rise in the use of 3D-printing and the unfortunate perspective of further pandemics should trigger the formulation of specific international standards dedicated to ventilation fittings.

## Material & methods

Five models of 3D-printed ventilation fittings were considered: F22, F22M22, M22M22, M22M22M22T, and F18F20 (Fig 1, Table 1). All designs except M22M22M22T were expected to produce intentional leaks.

Three different groups of 3D-printed devices were considered.

1. Devices manufactured in the central emergency 3D-printing platform of AP-HP using a F120 (Stratasys, Eden Prairie, USA) FDM printer: F22 (references 1–2), F22M22 (references 3–4), M22M22 (references 5–6), and M22M22M22T (references 7–8) produced by designers from BONE 3D (Paris, France) with black ABS-M30 (355–02112, Stratasys, Eden Prairie, USA) and soluble support SR30 (311–30200, Stratasys, Eden Prairie, USA);

2. FDM-printed F18F20 fittings (reference 9 –[6]) provided by Kernel Biomedical (https://3dleak.kernelbiomedical.com), printed with a Raise3D (Irvine, USA) machine using 1.75 mm Polylactic Acid (PLA) and 100 μm layers.

3. Devices manufactured at Necker–Enfants Malades University Hospital using a J735 Polyjet printer F120 (Stratasys, Eden Prairie, USA): F22M22 (reference 10) and M22M22 (reference 11), produced by designers from BONE 3D (Paris, France) with biocompatible

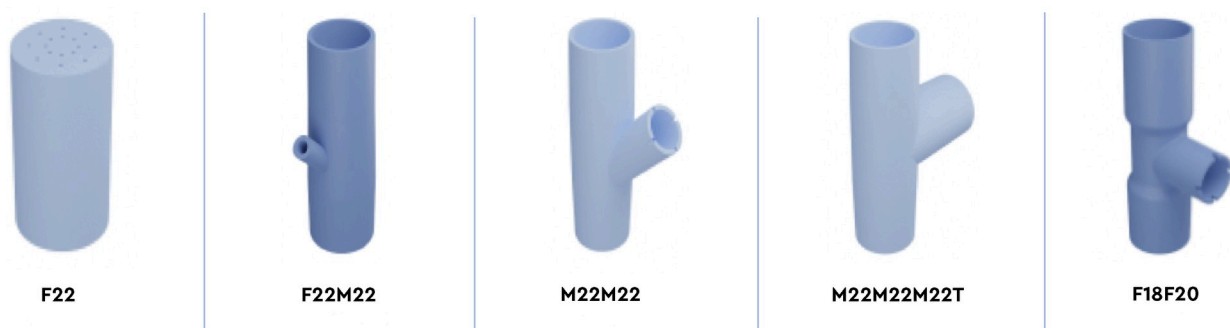

**Fig 1. Five models of 3D-printed ventilation fittings: F22 (references 1 and 2), F22M22 (references 4, 4, and 10), M22M22 (references 5, 6, and 11), M22M22M22T (references 7 and 8), and F18F20 (references 9).**

transparent MED610 (Stratasys, Eden Prairie, USA) resin and waterjet removable support SUP705 (Stratasys, Eden Prairie, USA) support.

Two parameters were considered for devices printed at the central AP-HP platform using FDM machines: (1) wall thickness and (2) coating. Thin walls were defined at 0.7 mm (with 0.18 mm layer thickness) and thick walls at 2.3 mm (with 0.33 layer thickness); 100% filling was used for all devices. Coating consisted in impregnating the printed fittings after post-processing using Nano Seal 180W+ (JELN Imprägnierung, Schwalmtal, Germany) to improve wall sealing (Table 1).

Five models of industrial leak valves–providing continuous leak paths in CPAP patient circuits when used with CPAP and bi-level machines–were considered as controls: (1) whisper swivel II exhalation port (332113, Philips, Amsterdam, Netherlands, reference 12), (2) disposable fixed exhalation port (DEP) with cap single-use (312149, Philips, Amsterdam, Netherlands, reference 13), (3) leak valve row fixed (24991, ResMed, San Diego, USA, reference 14), (4) WILAsilent swivel disposable exhalation port (1139909, WILAmed, Kammerstein, Germany, reference 15), and (5) Silentflow 2 exhalation system (WM23600, Lowenstein,

**Table 1. Printing characteristics and quality control for non-intentional leaks.**

| Reference | Design | Technique | Wall thickness (mm) | Coating (yes/no) | Quality control | | |
|---|---|---|---|---|---|---|---|
| | | | | | Leak at rigid junction (yes/no) | Leak at rubber junction (yes/no) | Parietal leak (yes/no) |
| 1 | F22 | FDM (ABS) | 0.7 | yes | yes | no | no |
| 2 | F22 | FDM (ABS) | 2.3 | yes | yes | no | no |
| 3 | F22M22 | FDM (ABS) | 0.7 | no | yes | no | yes |
| 4 | F22M22 | FDM (ABS) | 0.7 | yes | yes | no | no |
| 5 | M22M22 | FDM (ABS) | 0.7 | yes | yes | no | no |
| 6 | M22M22 | FDM (ABS) | 2.3 | yes | yes | no | no |
| 7 | M22M22M22T | FDM (ABS) | 2.3 | no | yes | no | yes |
| 8 | M22M22M22T | FDM (ABS) | 2.3 | yes | yes | no | no |
| 9 | F18F20 | FDM (PLA) | NA | no | yes | yes | yes |
| 10 | F22M22 | Polyjet (MED610) | NA | NA | no | no | no |
| 11 | M22M22 | Polyjet (MED610) | NA | NA | no | no | no |

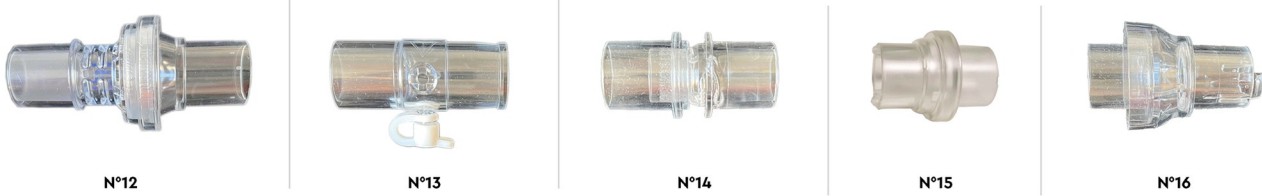

**Fig 2. Five models of industrial leak valves: (1)** whisper swivel II exhalation port (reference 12), **(2)** disposable fixed exhalation port (DEP) with cap single-use (reference 13), **(3)** leak valve row fixed (reference 14), **(4)** WILAsilent swivel disposable exhalation port (reference 15), and **(5)** Silentflow 2 exhalation system (reference 16).

Hamburg, Germany, reference 16) (Fig 2, Table 2).To screen for non-intentional leaks, we used an Astral 150 ventilator (ResMed, San Diego, USA) in double circuit with an adult profile and assist-control ventilation mode; 22 mm smoothbore tubes (Intersurgical, Wokingham, United Kingdom) were used to connect with the fittings. Rigid junctions when needed corresponded to Hudson RCI universal cuff connectors (41421 and 41422, TeleFlex Medical, Wayne, USA). Rubber junctions corresponded to F15F22 lipped elastomeric connectors (1701, Intersurgical, Wokingham, United Kingdom). Terminal obliterations of the fittings were obtained using 22F dust caps (1978000, Intersurgical, Wokingham, United Kingdom) and smaller caps (inner diameter: 8 mm) from zeolite molecular sieves of Inogen One G3 (Inogen, Goleta, USA) oxygen concentrators. Maximum pressure was set at 85 cmH$_2$O and fittings were immerged into water (see Discussion for the reference to the relevant regulatory texts). Leaks were screened for during five respiratory cycles for each fitting reference, with rigid and rubber junctions (Fig 3). All measures were performed twice: in May 2020, immediately after production, and in November 2021, 18 months after production.

In order to screen for intentional leaks, we used a Vivo 45 positive pressure generator (Breas Medical, Mölnlycke, Sweden) in constant mode without humidifier (expiratory pressure relief, ramp off) with a slim circuit (L327148, L3 Medical, Saint-Quentin-Fallavier, France), Hudson RCI universal cuff connectors (41421, TeleFlex Medical, Wayne, USA) and an air guard filter (1790000, Intersurgical, Wokingham, United Kingdom). A bacteria filter was connected between the ventilator and the breathing circuit. The leak valve was connected to the breathing circuit and a plug was placed in the mouth of the valve to measure the leakage during ventilation. Measurements were performed at five pressure values: 8, 10, 12, 14, and 16 cmH$_2$0 during five respiratory cycles (Figs 3 and 4).

All assessments were performed by ASV Santé (Genevilliers, France) by AML, KL, JPH, GB, RHK and DP. Data on printing time and costs were provided by BONE 3D (Paris, France).

**Table 2. Industrial leak valves used as controls.** Rendering adapted from images provided by the manufacturers.

| Reference | Description | Brand |
|---|---|---|
| Reference 12 | whisper swivel II exhalation port | Philips |
| Reference 13 | Disposable Fixed Exhalation Port (DEP) with Cap Single-use | Philips |
| Reference 14 | leak valve row fixed | Resmed |
| Reference 15 | WILAsilent swivel disposable exhalation port | WILAmed |
| Reference 16 | Silentflow 2 exhalation system | WILAmed |

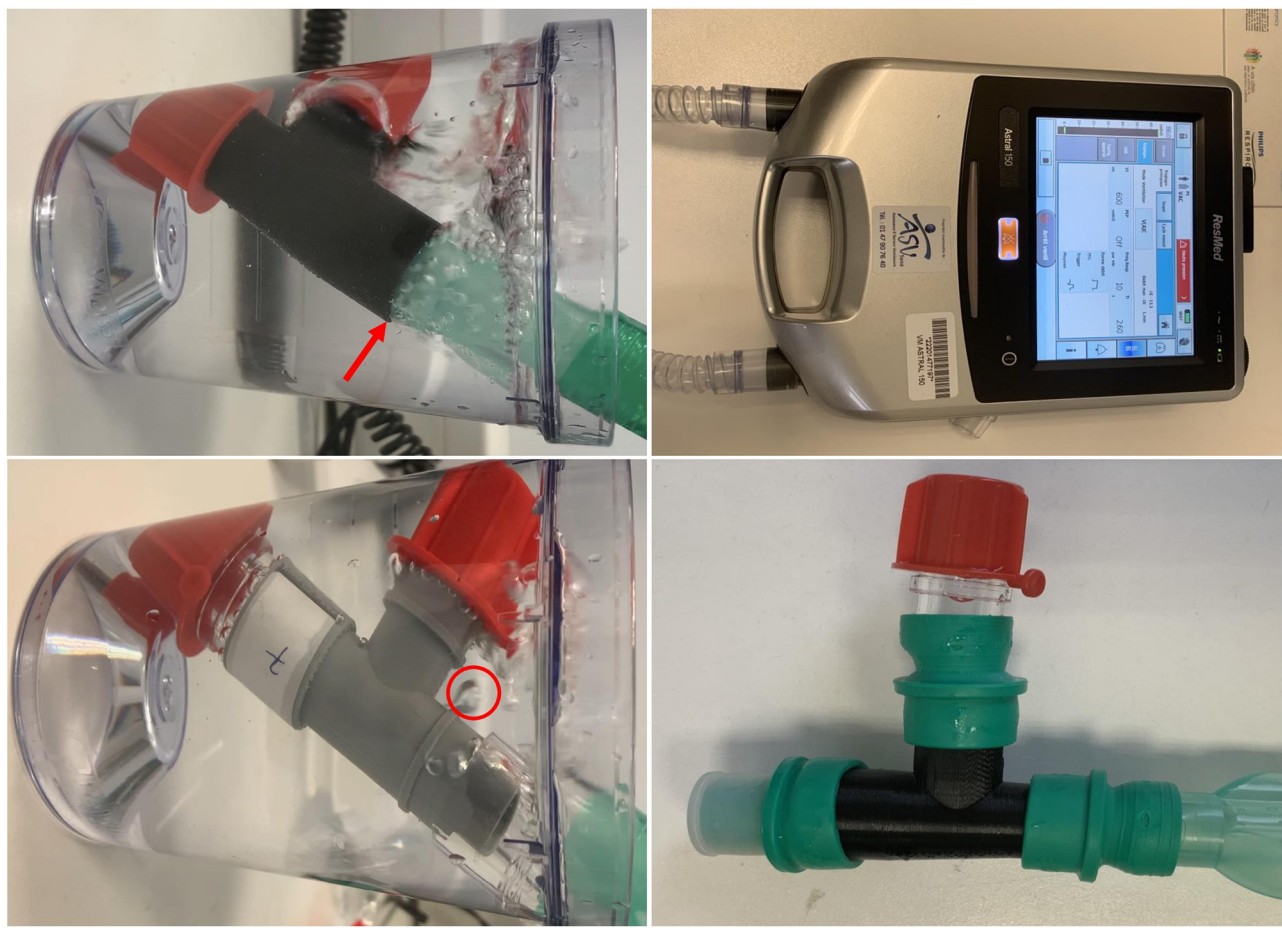

**Fig 3. (a)** Astral 150 ventilator (ResMed, San Diego, USA) in double circuit with an adult profile and assist-control ventilation mode used to assess non-intentional leaks; **(b)** M22M22M22T fitting (reference 7) with rubber junctions before being immerged into water for testing non-intentional leaks; **(c)** M22M22M22T fitting (reference 7) being tested for non-intentional leaks with rigid junctions, showing massive junctional leaks (red arrow); **(d)** F18F20 fitting (reference 9) being tested for non-intentional leaks with rigid junctions, showing significant junctional and parietal leaks (red circle indicated parietal leaks).

## Results

### 1. Non-intentional leaks (Table 1)

Industrial fittings (references 12–16) and Polyjet-printed fittings (references 10,11) had no leaks at rigid junctions, at rubber junctions, or parietal leaks. All FDM-printed fittings had leaks at rigid junctions (Fig 3), but this issue was tackled for the fittings printed within AP-HP (references 1–8) using rubber junctions. The FDM fitting printed by an external provider (reference 9) had persisting junctional leaks even with rubber junctions. All FDM-printed fittings (printed within AP-HP and from an external provider) had parietal leaks, even with thick walls (2.3 mm, reference 7), but this issue was tackled using coating (references 1,2,4,5,6,8), even for thin walls (0.7 mm, references 1,4,5). Close examination of devices after coating showed irregular junctional surfaces (Fig 5), potentially accounting for leaks when rigid junctions are used. Measures performed in May 2020 and in November 2021 showed identical results, supporting stable properties over time. The devices had been protected from light and stored in a medical office during this 18-months period.

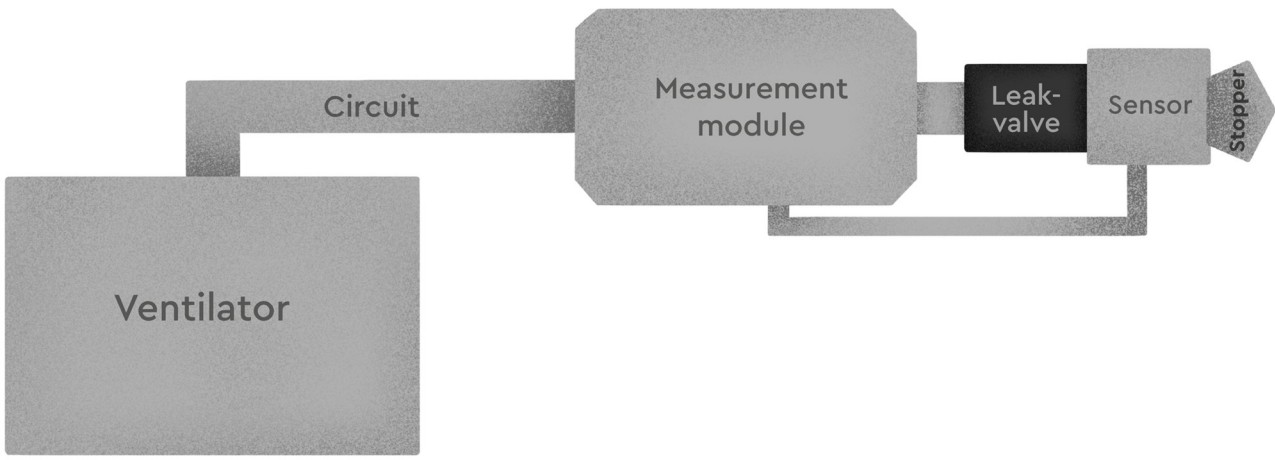

**Fig 4. Measurement system designed to assess intentional leaks.**

## 2. Intentional leaks (Tables 2 and 3)

All industrial (references 12–16) and Polyjet-printed (references 10–11) fittings had satisfactory intentional leaks. FDM-printed fittings without coating had acceptable levels of intentional leaks (references 3,9) but knowing that these two models had parietal leaks and should not be used in practice. Coating systematically blocked intentional leaks, due to the presence of coating material into the areas designed to allow leakage (Fig 5). In clear, the technical measures necessary to control unintentional parietal leaks in FDM-printed fittings blocked intentional leaks.

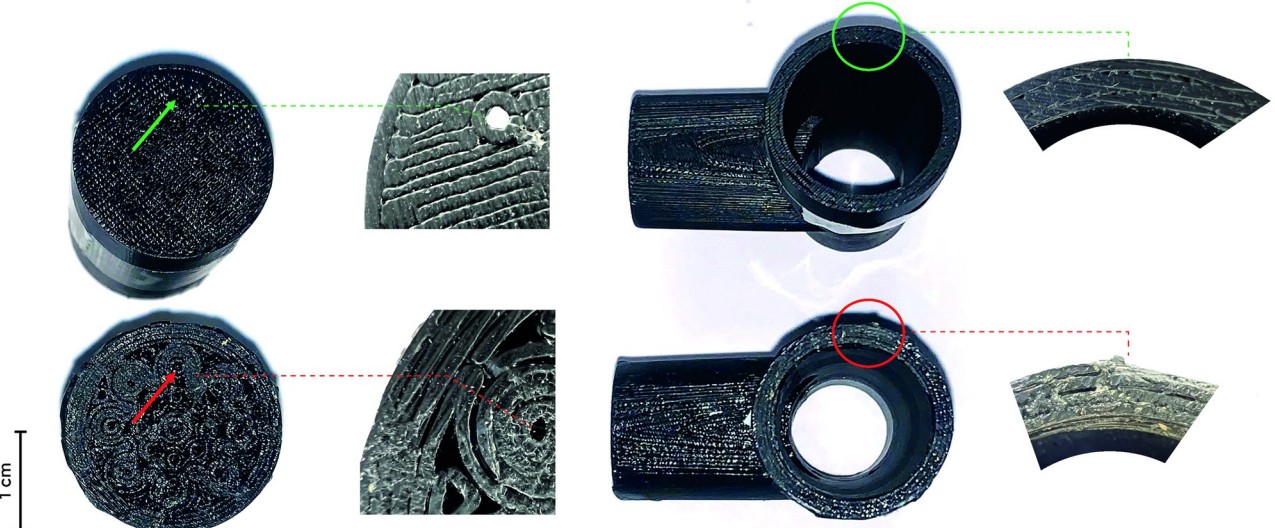

**Fig 5.** (a) Right panel: effects of coating on intentional leaks. Green arrow: reference 1 –F22 fitting, thin walls (0.7 mm), without coating, showing open perforations designed to allow intentional leaks. Red arrow: reference 1 –F22 fitting, thick walls (2.3 mm), with coating, showing obliterated perforations designed to allow intentional leaks. (b) Left panel: effects of coating on junctions. Green circle: reference 7 –M22M22M22T fitting, thick walls (2.3 mm), without coating showing clean surfaces at junction (green circle). Red circle: reference 8 –M22M22M22T fitting, thick walls (2.3 mm), with coating showing irregular surfaces at junction.

**Table 3. Quantification of intentional leaks at 5 increasing pressure levels.** References 7–8 have been excluded as T-fittings are not designed to produce intentional leaks. Bold characters: pressure values above 20 cmH$_2$O.

| | Design | Technique | Wall thickness (mm) | Coating | Pressure (cmH$_2$O) | | | | |
|---|---|---|---|---|---|---|---|---|---|
| | | | | | 8 | 10 | 12 | 14 | 16 |
| Reference 1 | F22 | FDM (ABS) | 0.7 | yes | 0–5 | 0–5 | 5–10 | 5–10 | 5–10 |
| Reference 2 | F22 | FDM (ABS) | 2.3 | yes | 0–5 | 0–5 | 0–5 | 0–5 | 0–5 |
| Reference 3 | F22M22 | FDM (ABS) | 0.7 | no | 15–20 | 15–20 | **20–25** | **20–25** | **20–25** |
| Reference 4 | F22M22 | FDM (ABS) | 0.7 | yes | 0–5 | 0–5 | 0–5 | 0–5 | 0–5 |
| Reference 5 | M22M22 | FDM (ABS) | 0.7 | yes | 0–5 | 0–5 | 0–5 | 0–5 | 0–5 |
| Reference 6 | M22M22 | FDM (ABS) | 2.3 | yes | 0–5 | 0–5 | 0–5 | 0–5 | 0–5 |
| Reference 9 | F18F20 | FDM (PLA) | 2.0 | no | **20–25** | **25–30** | **25–30** | **30–35** | **30–35** |
| Reference 10 | F22M22 | Polyjet (MED610) | 2.0 | NA | 15–20 | 15–20 | 15–20 | **20–25** | **20–25** |
| Reference 11 | M22M22 | Polyjet (MED610) | 2.0 | - | 5–10 | 5–10 | 5–10 | 5–10 | 10–15 |
| Reference 12 | - | industrial | - | - | 15–20 | **20–25** | **25–30** | **25–30** | **30–35** |
| Reference 13 | - | industrial | - | - | 15–20 | 15–20 | **20–25** | **20–25** | **20–25** |
| Reference 14 | - | industrial | - | - | **20–25** | **25–30** | **30–35** | **30–35** | **30–35** |
| Reference 15 | - | industrial | - | - | 15–20 | **20–25** | **20–25** | **25–30** | **30–35** |
| Reference 16 | - | industrial | - | - | 15–20 | **20–25** | **20–25** | **25–30** | **25–30** |

## 3. Production characteristics (Table 4)

FDM-printed devices were produced faster than Polyjet-printed devices, and were considerably cheaper, for equivalent designs.

## Discussion

Here we provide the first quality control assessment of an array of ventilation fittings produced during the first wave of the pandemic, from three different sources representing the suppliers of 3D-printed devices encountered by clinical departments during the crisis: (1) in-house dedicated emergency platforms, (2) in-house academic departments with previous 3D-printing

**Table 4. FDM and Polyjet printing time, requirements in raw materials and price.**

| Device | Time | Printing technique | Wall thickness (mm) | Resin | Support | Price |
|---|---|---|---|---|---|---|
| F22 (design 1) | 2h18 | FDM | 0.7 | ABS: 16,41 cm$^3$ | SR30: 1,35 cm$^3$ | 2,25 € |
| F22 (design 2) | 2h24 | FDM | 2.3 | ABS: 15,8 cm$^3$ | SR30: 1,35 cm$^3$ | 2,17 € |
| F22M22 (designs 3–4) | 3h40 | FDM | 0.7 | ABS: 21,9 cm$^3$ | SR30: 4,5 cm$^3$ | 3,36 € |
| M22M22 (design 5) | 3h24 | FDM | 0.7 | ABS: 22,8 cm$^3$ | SR30: 1,08 cm$^3$ | 3,03 € |
| M22M22 (design 6) | 3h24 | FDM | 2.3 | ABS: 21,59 cm$^3$ | SR30: 1,08 cm$^3$ | 2,87 € |
| M22M22M22T (designs 7–8) | 4h13 | FDM | 2.3 | ABS: 21,95 cm$^3$ | SR30: 9,12 cm$^3$ | 3,96 € |
| F22M22 (design 10) | 4h20 | Polyjet | 2.0 | MED610 (+ purge): 71g | SUP705: 30 g | 13,48 € |
| M22M22 (design 11) | 4h18 | Polyjet | 2.0 | MED610 (+ purge): 71 g | SUP705: 38 g | 14,31 € |

experience, and (3) independent external makers. We show that 3D-printing can be a valuable solution to overcome shortages but only under strict supervision.

Intentional leaks have a significant impact on the effectiveness of non-invasive ventilation as they prevent rebreathing of expired $CO_2$ [21] and should exceed 20 L/min when treating chronic respiratory failure and 22 L/min for acute respiratory failure [21,22]. We found average intentional leakage for all 3D models considered together at 13.5 ± 10.5 L/min, and at 25.8 ±5.0 L/min for control industrial models; this difference most probably reflects the effects of coating on the mechanical properties of FDM-printed devices: while coating seems mandatory to prevent parietal leakage, it interferes with intentional leaks and makes the devices unfit for clinical use (Fig 5).

Using ventilation devices with unintentional or insufficient intentional leaks can have severe clinical consequences, with detrimental effect on the effectiveness of NIV in acute and chronic conditions [23–26]. Similarly, leaks can aggravate nocturnal and diurnal hypoventilation [23–26] and contaminate the environment of the patient by diffusing viral particles [3,4]. This point is particularly relevant knowing that coating for FDM, although mandatory for preventing parietal leaks, leads to irregular surface at junctions, interfering with sealing and leading to unintentional leaks if rubber appliances are not used (Fig 5).

Our results stress the importance of professional 3D-printing protocols for producing critical devices such as ventilation fittings. In usual situations, the manufacture of such devices is subjected to the recent EU regulation 2017/745 (https://eur-lex.europa.eu/eli/reg/2017/745/oj), which limits emergency production due to demanding quality control and risk management protocols. To the best of our knowledge, these regulatory concerns were frequently mentioned but rarely addressed formally by the numerous teams who have produced 3D-printed ventilation fittings and other respiratory-related devices during the first wave of the pandemic [27–30], and few initiatives have been successfully certified by local regulatory authorities [31,32]. If further sanitary crises occur, in potential cases of temporary adaptations of this regulation to overcome shortages, our results strongly suggest that strict quality control assessments should be maintained, managed by teams experienced in medical 3D-printing, to eventually obtain formal certification. Choices in printing methods, such as Polyjet for instance, will depend on financial considerations (Table 4), on the background of the 3D-printing engineering teams, and on the volume of material required, including considerations on the conservation and rate of use of the printed devices [33–39].

Interestingly, the methods for testing leaks of ventilation fittings are not codified by current international standards such as (1) EN 12342:1998+A1:2009 Breathing tubes intended for use with anaesthetic apparatus and ventilators, and (2) EN 13544–2:2002+A1:2009 Respiratory therapy equipment—Part 2: Tubing and connectors. The assessment method we used, although basic, provides clear answers to the clinical issues raised using 3D-printed fittings: occurrence of parietal leaks, occurrence of junctional leaks, and efficiency of intentional leaks. High pressures (85 cmH$_2$O) used to evaluate non-intentional leaks may come out as extreme, especially regarding CPAP standards. However, such pressures may occur in the events such as coughing, and devices should remain sealed in situations with risks of viral particle diffusion. Beyond the current concerns related to the pandemic, our study underlines the need for a standardization of the quality control methods for ventilation tubes, knowing the current rise in the production of 3D-printed devices and specific risks caused by poorly designed and produced fittings.

## Conclusion

Additive manufacturing is a valid technique for producing ventilation devices such as fittings with intentional leaks. Our results showed that both FDM with coating and Polyjet allow to

obtain devices without parietal leaks. We also demonstrated that these physical properties are stable in time, at least for 18 months without exposition to light, supporting the perspective of the constitution of potential stocks of 3D-printed ventilation devices. FDM with coating nevertheless impaired several of the main properties of the fitting with intentional leaks by creating irregular junctions and obliterating the zones designed to lead intentionally. Based on these findings, our work strongly indicates that professional supervision is mandatory to choose the most relevant production technique, based on technical requirements and local financial constraints. FDM, although cheap and dependable, is not a straightforward approach for producing devices with intentional leaks. Polyjet seems to fulfil most requirements but is not easily available to healthcare professionals and is still expensive. We furthermore suggest that there is a critical need both for (1) defining clear protocols for 3D-printing emergency devices in case of further situations of shortage, and (2) standardizing assessment methods specifically dedicated to the quality control of ventilation fittings. Safety concerns should remain at the forefront, even during sanitary crises: while current regulations are not compatible with fast-track certification for emergency 3D-printing, the use of this technology in extreme situations in the future will only be conceivable based on the formulation of fast-track but reliable assessment methods for ventilation fittings.

## Supporting information

**S1 Design. STL file for references 1–2 (F22).**
(STL)

**S2 Design. STL file for references 3–4 (F22M22).**
(STL)

**S3 Design. STL file for references 5–6 (M22M22).**
(STL)

**S4 Design. STL file for references 7–8 (M22M22M22).**
(STL)

**S1 Video. Assessing leaks in reference 6: FDM-printed M22M22 with thick (2.3 mm) walls and coating, tested using rubber junctions.** Absence of junctional or parietal leaks.
(MOV)

**S2 Video. Assessing leaks in reference 11: Polyjet-printed M22M22, tested using rigid junctions.** Absence of junctional or parietal leaks.
(MOV)

**S3 Video. Assessing leaks in reference 9: FDM-printed F18F20, tested using rigid junctions.** Multiple junctional and parietal leaks.
(MOV)

## Acknowledgments

The authors thank the manufacturers for supplying industrial ventilation fitttings used for assessment. Special thanks to AVS Santé (Genevilliers, France) for the time dedicated to leak assessment, and to Emma Blanc-Tailleur (École Estienne) for illustrations.

## Author Contributions

**Conceptualization:** Roman Hossein Khonsari, Mathilde Oranger, Pierre-Marc François, Alexis Mendoza-Ruiz, Capucine Morelot-Panzini, Maxime Patout, Jésus Gonzalez-Bermejo.

**Data curation:** Roman Hossein Khonsari, Pierre-Marc François, Alexis Mendoza-Ruiz, Karl Leroux, Ghilas Boussaid, Delphine Prieur, Jean-Pierre Hodge, Antoine Belle, Maxime Patout.

**Formal analysis:** Roman Hossein Khonsari, Mathilde Oranger, Pierre-Marc François, Ghilas Boussaid, Jean-Pierre Hodge, Antoine Belle, Vincent Midler, Capucine Morelot-Panzini, Jésus Gonzalez-Bermejo.

**Methodology:** Roman Hossein Khonsari, Alexis Mendoza-Ruiz, Karl Leroux, Vincent Midler.

**Visualization:** Roman Hossein Khonsari, Maxime Patout, Jésus Gonzalez-Bermejo.

**Writing – original draft:** Roman Hossein Khonsari, Pierre-Marc François, Delphine Prieur, Jésus Gonzalez-Bermejo.

**Writing – review & editing:** Roman Hossein Khonsari, Jésus Gonzalez-Bermejo.

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
