## [Decision Letter · Decision Letter 0]

4 Mar 2022

PONE-D-22-02617Quality versus emergency: how good were ventilation fittings produced by additive manufacturing to address shortages during the COVID19 pandemic?PLOS ONE

Dear Dr. Khonsari,

Thank you for submitting your manuscript to PLOS ONE. After careful consideration, we feel that it has merit but does not fully meet PLOS ONE’s publication criteria as it currently stands. Therefore, we invite you to submit a revised version of the manuscript that addresses the points raised during the review process.

We look forward to receiving your revised manuscript.

Kind regards,

Mohammad Azadi

Academic Editor

PLOS ONE

Journal Requirements:

Additional Editor Comments:

About suggested references by the reviewer, you could decide either to cite them or not, if they are not related to your topic. Moreover, all abbreviations need to be defined at first mentioning. The introduction should be extended and the novelty should be highlighted compared to the literature review, if there is any. The conclusion part could be rewritten one by one to highlight the obtained results. The number of references should be extended to 35 articles for a better literature review and also a better discussion. The scale bar should be added on images to know the size of samples. A microscopic image should be added besides Figure 3 to show the quality of samples. In Table 3, what is the dimension of the wall thickness? It is better to add a table for different parameters of 3D-printing for all specimens.

Reviewers' comments:

Reviewer's Responses to Questions

**Comments to the Author**

1. Is the manuscript technically sound, and do the data support the conclusions?

Reviewer #1: Yes

Reviewer #2: Yes

2. Has the statistical analysis been performed appropriately and rigorously? 

Reviewer #1: No

Reviewer #2: Yes

3. Have the authors made all data underlying the findings in their manuscript fully available?

Reviewer #1: Yes

Reviewer #2: Yes

4. Is the manuscript presented in an intelligible fashion and written in standard English?

Reviewer #1: Yes

Reviewer #2: Yes

5. Review Comments to the Author

Reviewer #1: Tables 1-3 are missing. So, it is not possible to follow the discussions made by the authors.

The reasons for using ABS for this particular application should be described.

The authors employed five models of industrial leak valves as controls. What are their specifications? What are the essential differences? Summarize this information in a table.

Provide a drawing of the ventilation fitting.

Reviewer #2: The additive manufacturing (AM) or 3D printing technique becomes a supplementary manufacturing process to meet the explosive demands of essential medical equipments and to ease the health disaster worldwide during COVID-19 pandemic in the last two years. In this manuscript, authors provide key elements towards a better quality control of 3D-printed ventilation fittings in a context of sanitary crisis using two different 3D techniques. The investigation is timely, worth for investigation, well-structured and easy to follow. Hence, I would recommend its acceptance. However, authors must incorporate some modifications:

1. In page 6, starting the line 'we considered (1) in house devices ..... AP-HP 3D- printing platform using FDM and ABS as a raw material,..", please correct to..... ".... using FDM printing technique in which ABS is used as a raw material".

Similarly, please correct.... "(2) in house devices produced........ using polyjet and MED610 as a raw material,............... to... "".... using PolyJet printing technique whereas MED610 is used as a raw material".

2. PolyJet is a printing technique. Replace polyjet everywhere in the manuscript by "PolyJet'

3. Please note that many 'articles' are missing in the manuscript, e.g. in Page 6... "......MED610 (Stratasys, Eden Prairie, USA) resin as raw material. " will be."......MED610 (Stratasys, Eden Prairie, USA) resin as a raw material. " (a is missing here

4. Finally, I must say that Literature section of the manuscript is very poor. Please note that there are several excellent publications appeared in the last one year in 3D printing and COVID19 crisis. Few more references must be mentioned here, e.g.,

Tareq et al. https://www.sciencedirect.com/science/article/pii/S0278612520302351

Andres et al. https://www.sciencedirect.com/science/article/pii/S0278612521000716

Hannah et al. https://www.sciencedirect.com/science/article/pii/S0278612521001473

Once authors revise the manuscript taking my aforementioned points into account, I will be happy to accept it.

6. PLOS authors have the option to publish the peer review history of their article (what does this mean?). If published, this will include your full peer review and any attached files.

Reviewer #1: No

Reviewer #2: No

---

## [Author Response · Author response to Decision Letter 0]

8 Mar 2022

Dear Editors of PLoS ONE,

Many thanks for your positive comments on our manuscript.

Please find in the following the requirements of the Academic Editor and of the reviewers (in bold characters) followed by our answers (in italics).

We hope that our manuscript, in its present form, will be more suitable for publication.

Thanks in advance for the time you will spend considering our re-submission.

Best regards,

RH Khonsari, for the co-authors

A. Academic editor

https://journals.plos.org/plosone/s/file?id=wjVg/PLOSOne_formatting_sample_main_body.pdf
https://journals.plos.org/plosone/s/file?id=ba62/PLOSOne_formatting_sample_title_authors_affiliations.pdf

Style requirements have been checked.

Tables have been included into the main manuscript.

3. Please ensure that you have an ORCID iD and that it is validated in Editorial Manager.

ORCID ID has been provided.

4. About suggested references by the reviewer, you could decide either to cite them or not if they are not related to your topic. Moreover, all abbreviations need to be defined at first mentioning. The introduction should be extended, and the novelty should be highlighted compared to the literature review, if there is any. The conclusion part could be rewritten one by one to highlight the obtained results. The number of references should be extended to 35 articles for a better literature review and a better discussion.

These modifications have been performed, and the references have been extended to less that 35 articles, but including most of the recent relevant studies on the subject.

5. The scale bar should be added on images to know the size of samples. A microscopic image should be added besides Figure 3 to show the quality of samples. In Table 3, what is the dimension of the wall thickness? It is better to add a table for different parameters of 3D-printing for all specimens.

Required modifications to figures and tables have been performed. Close views of the devices in Figure 3 have been added.

B. Reviewer №1

1. Tables 1-3 are missing. So, it is not possible to follow the discussions made by the authors.

Tables 1-3 are now inserted into the main text.

2. The reasons for using ABS for this application should be described.

The authors employed five models of industrial leak valves as controls. What are their specifications? What are the essential differences? Summarize this information in a table. Provide a drawing of the ventilation fitting.

All these technical data have been added. The drawings of the ventilation fittings are provided in the tables.

C. Reviewer №2

1. In page 6, starting the line 'we considered (1) in house devices ..... AP-HP 3D- printing platform using FDM and ABS as a raw material,..", please correct to..... ".... using FDM printing technique in which ABS is used as a raw material".

This modification has been performed.

2. Similarly, please correct.... "(2) in house devices produced........ using polyjet and MED610 as a raw material,............... to... "".... using PolyJet printing technique whereas MED610 is used as a raw material".

This modification has been performed.

3. PolyJet is a printing technique. Replace polyjet everywhere in the manuscript by "PolyJet'

This modification has been performed.

4. Please note that many 'articles' are missing in the manuscript, e.g. in Page 6... "......MED610 (Stratasys, Eden Prairie, USA) resin as raw material. " will be."......MED610 (Stratasys, Eden Prairie, USA) resin as a raw material. " (a is missing here

This modification has been performed.

5. Finally, I must say that Literature section of the manuscript is very poor. Please note that there are several excellent publications appeared in the last one year in 3D printing and COVID19 crisis. Few more references must be mentioned here, e.g.,

Tareq et al. https://www.sciencedirect.com/science/article/pii/S0278612520302351

Andres et al. https://www.sciencedirect.com/science/article/pii/S0278612521000716

Hannah et al. https://www.sciencedirect.com/science/article/pii/S0278612521001473

More references (including references provided by Reviewer №2 have been added.

---

## [Editor Report · Decision Letter 1]

10 Mar 2022

PONE-D-22-02617R1Quality versus emergency: how good were ventilation fittings produced by additive manufacturing to address shortages during the COVID19 pandemic?PLOS ONE

Dear Dr. Khonsari,

Thank you for submitting your manuscript to PLOS ONE. After careful consideration, we feel that it has merit but does not fully meet PLOS ONE’s publication criteria as it currently stands. Therefore, we invite you to submit a revised version of the manuscript that addresses the points raised during the review process.

We look forward to receiving your revised manuscript.

Kind regards,

Mohammad Azadi

Academic Editor

PLOS ONE

Journal Requirements:

Additional Editor Comments (if provided):

The revised article is not accepted and it should be carefully revised again. All changes should be highlighted in the revised article. Please be careful to address all comments. One example of not addressing the comment is to rewritten the conclusion part, one by one, to show the novelty. This issue was not done in the revision by authors. So please check all comments again, one by one, carefully. Moreover, mentioning "requirement modifications were performed" is not enough. More details should be mentioned and changes should be highlighted in the revised article. In addition, the scale bar in Figure 3 is too small and it could be read. The number of references is not extended and etc. 

---

## [Author Response · Author response to Decision Letter 1]

13 Mar 2022

Dear editors of PLoS ONE,

Many thanks on your insightful comments on our manuscript. Please find in the following the list of the modifications required by the academic editor and the reviewers (in bold characters) followed by our answers (in italics).

We hope that our manuscript, in its present form, will be closer to your expectations.

Thanks in advance for the time you will spend considering our revisions.

Best regards,

RH Khonsari, for the co-authors.

A. Academic editor

Style requirements have been checked.

Tables have been included into the main manuscript.

3. Please ensure that you have an ORCID iD and that it is validated in Editorial Manager.

ORCID ID has been provided.

4. About suggested references by the reviewer, you could decide either to cite them or not if they are not related to your topic. Moreover, all abbreviations need to be defined at first mentioning. The introduction should be extended, and the novelty should be highlighted compared to the literature review, if there is any. The conclusion part could be rewritten one by one to highlight the obtained results. The number of references should be extended to 35 articles for a better literature review and a better discussion.

All abbreviations (such as FDM for instance) have been defined when first used. The introduction has been extended with the addition of two paragraphs: one on the importance of regulation issues and another one on the technical choices made by our team (choice of FDM, choice of ABS). The conclusion has been also extended and now includes a list of the main points raised by this paper: (1) potentialities of 3D printing in producing ventilation devices during crises, (2) need for technical skills in the production of medical devices, (3) need for support from the regulation bodies, and (4) need for a clarification of assessment panels, especially for fittings with intentional leaks. The references have furthermore been extended to more than 35 articles, and now including most of the recent relevant studies on the subject.

5. The scale bar should be added on images to know the size of samples. A microscopic image should be added besides Figure 3 to show the quality of samples. In Table 3, what is the dimension of the wall thickness? It is better to add a table for different parameters of 3D-printing for all specimens.

Required modifications to figures and tables have been performed. A new table has been added with the characteristics of the commercial devices we included into the assessment. Close views of the devices in Figure 3 have been added and a larger scale bar (1 cm) has been included.

B. Reviewer №1

1. Tables 1-3 are missing. So, it is not possible to follow the discussions made by the authors.

Tables 1-3 are now inserted into the main text.

2. The reasons for using ABS for this application should be described.

The authors employed five models of industrial leak valves as controls. What are their specifications? What are the essential differences? Summarize this information in a table. Provide a drawing of the ventilation fitting.

All these technical data have been added in the introduction. The drawings of the ventilation fittings are provided in a new table dedicated to the commercial fittings we included into the assessment (also following a requirement of the academic editor).

C. Reviewer №2

1. In page 6, starting the line 'we considered (1) in house devices ..... AP-HP 3D- printing platform using FDM and ABS as a raw material,..", please correct to..... ".... using FDM printing technique in which ABS is used as a raw material".

This modification has been performed.

2. Similarly, please correct.... "(2) in house devices produced........ using polyjet and MED610 as a raw material,............... to... "".... using PolyJet printing technique whereas MED610 is used as a raw material".

This modification has been performed.

3. PolyJet is a printing technique. Replace polyjet everywhere in the manuscript by "PolyJet'

This modification has been performed.

4. Please note that many 'articles' are missing in the manuscript, e.g. in Page 6... "......MED610 (Stratasys, Eden Prairie, USA) resin as raw material. " will be."......MED610 (Stratasys, Eden Prairie, USA) resin as a raw material. " (a is missing here

This modification has been performed.

5. Finally, I must say that Literature section of the manuscript is very poor. Please note that there are several excellent publications appeared in the last one year in 3D printing and COVID19 crisis. Few more references must be mentioned here, e.g.,

Tareq et al. https://www.sciencedirect.com/science/article/pii/S0278612520302351

Andres et al. https://www.sciencedirect.com/science/article/pii/S0278612521000716

Hannah et al. https://www.sciencedirect.com/science/article/pii/S0278612521001473

More references, including references above, have been added, also following a requirement of the academic editor.

---

## [Decision Letter · Decision Letter 2]

30 Mar 2022

Quality versus emergency: how good were ventilation fittings produced by additive manufacturing to address shortages during the COVID19 pandemic?

PONE-D-22-02617R2

Dear Dr. Khonsari,

We’re pleased to inform you that your manuscript has been judged scientifically suitable for publication and will be formally accepted for publication once it meets all outstanding technical requirements.

Kind regards,

Mohammad Azadi

Academic Editor

PLOS ONE

Additional Editor Comments (optional):

It is accepted based on the reviewers' comments. Only please remove the reference in the conclusion part when the proof file will be sent to you. It should be moved to the discussion part.

Reviewers' comments:

Reviewer's Responses to Questions

**Comments to the Author**

1. If the authors have adequately addressed your comments raised in a previous round of review and you feel that this manuscript is now acceptable for publication, you may indicate that here to bypass the “Comments to the Author” section, enter your conflict of interest statement in the “Confidential to Editor” section, and submit your "Accept" recommendation.

Reviewer #1: All comments have been addressed

Reviewer #2: All comments have been addressed

2. Is the manuscript technically sound, and do the data support the conclusions?

Reviewer #1: Yes

Reviewer #2: Yes

3. Has the statistical analysis been performed appropriately and rigorously? 

Reviewer #1: Yes

Reviewer #2: Yes

4. Have the authors made all data underlying the findings in their manuscript fully available?

Reviewer #1: Yes

Reviewer #2: (No Response)

5. Is the manuscript presented in an intelligible fashion and written in standard English?

Reviewer #1: Yes

Reviewer #2: Yes

6. Review Comments to the Author

Reviewer #1: (No Response)

Reviewer #2: Authors took all of my comments into account and improved the manuscript significantly. Hence, I am happy to accept it now.

7. PLOS authors have the option to publish the peer review history of their article (what does this mean?). If published, this will include your full peer review and any attached files.

Reviewer #1: No

Reviewer #2: No

---

## [Editor Report · Acceptance letter]

13 Apr 2022

PONE-D-22-02617R2 

Quality versus emergency: how good were ventilation fittings produced by additive manufacturing to address shortages during the COVID19 pandemic? 

Dear Dr. Khonsari:

I'm pleased to inform you that your manuscript has been deemed suitable for publication in PLOS ONE. Congratulations! Your manuscript is now with our production department. 

Kind regards, 

on behalf of

Dr. Mohammad Azadi 

Academic Editor

PLOS ONE